# Antiplasmodial Cyclodecapeptides from Tyrothricin Share a Target with Chloroquine

**DOI:** 10.3390/antibiotics11060801

**Published:** 2022-06-14

**Authors:** Adrienne N.-N. Leussa, Marina Rautenbach

**Affiliations:** BioPep Peptide Group, Department of Biochemistry, University of Stellenbosch, Stellenbosch 7600, South Africa; adrienne.leussa@54gene.com

**Keywords:** antimicrobial peptides, cyclic peptides, tyrocidine, tyrothricin, antimalarial activity, malaria, *Plasmodium falciparum*, chloroquine resistance

## Abstract

Previous research found that the six major cyclodecapeptides from the tyrothricin complex, produced by *Brevibacillus parabrevis*, showed potent activity against chloroquine sensitive (CQS) *Plasmodium falciparum*. The identity of the aromatic residues in the aromatic dipeptide unit in cyclo-(D-Phe^1^-Pro^2^-(Phe^3^/Trp^3^)-D-Phe^4^/D-Trp^4^)-Asn^5^-Gln^6^-(Tyr^7^/Phe^7^/Trp^7^)-Val^8^-(Orn^9^/Lys^9^)-Leu^10^ was proposed to have an important role in activity. CQS and resistant (CQR) *P. falciparum* strains were challenged with three representative cyclodecapeptides. Our results confirmed that cyclodecapeptides from tyrothricin had significantly higher antiplasmodial activity than the analogous gramicidin S, rivaling that of CQ. However, the previously hypothesized size and hydrophobicity dependent activity for these peptides did not hold true for *P. falciparum* strains, other than for the CQS 3D7 strain. The Tyr^7^ in tyrocidine A (TrcA) with Phe^3^-D-Phe^4^ seem to be related with loss in activity correlating with CQ antagonism and resistance, indicating a shared target and/or resistance mechanism in which the phenolic groups play a role. Phe^7^ in phenycidine A, the second peptide containing Phe^3^-D-Phe^4^, also showed CQ antagonism. Conversely, Trp^7^ in tryptocidine C (TpcC) with Trp^3^-D-Trp^4^ showed improved peptide selectivity and activity towards the more resistant strains, without overt antagonism towards CQ. However, TpcC lead to similar parasite stage inhibition and parasite morphology changes than previously observed for TrcA. The disorganization of chromatin packing and neutral lipid structures, combined with amorphous hemozoin crystals, could account for halted growth in late trophozoite/early schizont stage and the nanomolar non-lytic activity of these peptides. These targets related to CQ antagonism, changes in neural lipid distribution, leading to hemozoin malformation, indicate that the tyrothricin cyclodecapeptides and CQ share a target in the malaria parasite. The differing activities of these cyclic peptides towards CQS and CQR *P. falciparum* strains could be due to variable target interaction in multiple modes of activity. This indicated that the cyclodecapeptide activity and parasite resistance response depended on the aromatic residues in positions 3, 4 and 7. This new insight on these natural cyclic decapeptides could also benefit the design of unique small peptidomimetics in which activity and resistance can be modulated.

## 1. Introduction

Among the diseases that occur in tropical countries, malaria is one of the most frequent causing approximately 228 million new infections and an upper estimate of 738,000 deaths worldwide during the first year (2020) of the COVID-19 pandemic, with nearly 95% of the malaria burden and deaths occurring in sub-Saharan Africa [1]. This death toll in one year is more than triple that over two years of the COVID-19 pandemic in Africa [2]. Most malaria cases are caused by the unicellular protozoan and obligate intracellular parasite *Plasmodium falciparum*. Therefore, *P. falciparum* derived malaria is presently still a public health problem especially in developing countries. Malaria control is part of the sixth Millennium Development Goal (MDG), Target 6.C—“to have halted by 2015 and begun to reverse the incidence of malaria and other major diseases” [1]. Now, seven years later the resistance by the parasites to antimalarial drugs is still the major drawback to the control of malaria and a major public health concern in developing and third world countries. For example, chloroquine (CQ) that used to be the first line antimalarial drug because of its low cost, low incidence of side-effects and high efficacy [3] has since been abandoned because of an upsurge of CQ resistant (CQR) *P. falciparum* [4]. The lurking threat of resistance by these parasites to the artemisinin, the major active in current antimalarial drugs (artemisinin combination therapies or ACTs), signals the urgency to develop new drugs that have novel targets in *P. falciparum* [5,6,7,8,9].

Among the approaches to develop new antimalarial drugs, the modified erythrocyte membrane following infection by the malaria parasite is exploited as a target [10,11,12,13,14]. Some of the changes that occur in the infected erythrocyte plasma membrane (IEPM) include increased fluidity [13,15,16,17], novel pores that boost membrane permeability [18,19,20,21,22], transfer of the anionic phosphatidylserine (PS) from the inner leaflet to the outer leaflet of the bilayer which changes the membrane lipid asymmetry and the infected erythrocyte tends to be similarly anionic like bacterial cells [10,23,24]. These erythrocyte membrane changes increase the likelihood that membrane active compounds like cationic antimicrobial peptides (AMPs) and membrane active antibiotics may have selective activity towards the infected erythrocyte membrane [25,26] as well as the parasitophorous vacuole membrane that is derived from the host cell [14]. Indeed, AMPs have been investigated as a probable source of future antimalarial drugs due to some valuable properties including those considered for their use as new antimicrobials [27,28,29,30]. Though originating from various sources, the structural characteristics of AMP families allows for the fine-tuning of their physicochemical characteristics to improve their antimicrobial properties using structure–activity relationship (SAR) studies. Using the effect of parameters such as size, charge, amphipathicity/hydrophobicity, supramolecular organization and conformational flexibility on the interaction with microbial targets, AMP structure and activity can be optimized [27].

One of the first studies using peptides as antimalarials was on the tyrothricin complex produced by *Brevibacillus parabrevis*, and this complex showed comparable activity to quinine towards *P. gallinaceum* in infected chickens [28]. Gramicidins, linear pentadecapeptides from tyrothricin were shown to have potent selective anti-*Plasmodium* activity related to K^+^ active channels in the infected erythrocyte [29,30]. However, cyclic peptides have better potential as therapeutics because and enhanced stability towards proteolytic degradation leading to better bioavailability, as well as improved receptor selectivity due to a more rigid conformation [31]. Among cyclic AMPs with potent and selective anti-*Plasmodium* activity, most are secondary metabolites from bacteria and fungi with a few examples given below. Aerucyclamide B from the cyanobacterium *Microcystis aeruginosa* showed selective sub-micromolar activity again the blood stages of *P. falciparum* [32]. Cyclic octadepsipeptides, octaminomycins A and B, isolated from a *Streptomyces* species, exhibited low micromolar selective activity against a CQS strain, as well as two CQR strains of *P. falciparum* [33]. Cyclomarin A, a cyclic hexapeptide from a marine streptomycete is as a selective growth inhibitor of *P. falciparum* in the nanomolar range with *P. falciparum* diadenosine triphosphate hydrolase (PfAp3Aase) as target [34]. Cyclic decapeptides from the tyrocidine group (Figure 1) are also produced as part of the antiplasmodial tyrothricin complex along with linear gramicidins [35]. Rautenbach et al. [36] revealed the potent sub-nanomolar antiplasmodial activity of the six major tyrocidines from tyrothricin, as a consequence of inhibition of the development and life cycle progress of CQS *P. falciparum* 3D7.

The membrane active cyclic peptide GS (Figure 1) from *Aneurinibacillus migulanus* [37], share the conserved VOLfP pentapeptide with the Trcs and is also antiplasmodial, although its activity was significantly lower in comparison to the Trcs [36]. This indicated that the variable pentapeptide in the Trcs have a major influence on the anti-*Plasmodium* activity. However, some inherent systemic toxicity of the Trcs still remain, which Jokonya et al. [38] addressed in a “smart” pH triggered nano-conjugate containing the Trcs linked to poly(N-vinylpyrrolidone). In this study, the hemolytic toxicity was lowered by nearly 20-fold in the nano-formulation while a low nanomolar activity was maintained against a CQS *P. falciparum* strain.

Nevertheless, these two studies only evaluated Trc activity towards CQS *P. falciparum* strains and considered only the six major Trcs in commercial tyrothricin. The SAR analyses from Rautenbach et al. [36] deduced that the anti-*P. falciparum* activity of the (Trcs) tyrocidines and selectivity for the infected erythrocyte or parasite was determined by overall hydrophobicity and steric factors. The D-residue in the second position of the variable XxNQY pentapeptide unit was proposed to be essential in the activity, with the most hydrophobic and smallest peptide, TrcA (Figure 1), being the most active and the largest most polar peptide, tyrocidine C (TrcC, cyclo(fPWwNQYVOL), the least active [36]. In this study, we challenged the SAR hypothesis, comparing the activity of PhcA (Figure 1), a smaller more hydrophobic Phe-rich TrcA analogue with D-Phe^4^ and a larger more polar Trp-rich TrcC analogue with D-Trp^4^, TpcC (Figure 1), with that of TrcA against three *P. falciparum* strains with variable CQ sensitivity/resistance. As we found that the activity of the three peptides were not maintained against the three strains, indicating a possible relationship with CQR, the study was expanded to check their influence when combined with CQ. These results also led to a high-resolution fluorescence microcopy study on the effect of TpcC on the parasite morphology, in particular assessing neutral lipids as a possible target, as it is important for hemozoin formation and CQ activity.

## 2. Results

### 2.1. P. falciparum Strain Susceptibility and Cytotoxicity of TrcA, PhcA and TpcC

Previously Rautenbach et al. [36] observed that the natural Trc mixture, individual purified Trc analogues, and specifically TrcA, had significantly higher antiplasmodial activity than the analogous GS, with IC_50_’s ranging between 0.6 and 460 nM against the CQS *P. falciparum* 3D7 strain. Similarly, Jokonya et al. [38] found that the Trc mixture had an IC_50_ of 1.1 nM against the CQS *P. falciparum* NF54 strain.

In this study, we also found low nanomolar IC_50_’s of 41 nM and 23 nM for the more hydrophobic TrcA and Phe-rich PhcA towards the CQS 3D7 strain, with a much higher IC_50_ of 126 nM by the more polar Trp-rich TpcC (Table 1). This correlated with the decrease in activity with increase in polarity and size found by Rautenbach et al. [36]. However, this trend was not followed when the three cyclic decapeptides were tested against the *P. falciparum* D10 strain. We classified the D10 strain in our culture library as have intermittent CQ resistance (CQI) because of a significant (*p* = 0.0039) increase the QC IC_50_ against the D10 strain compared to the 3D7 strain (Table 1). TrcA was the least active against the D10 stain but the difference in the IC_50_ towards the CQI D10 strain and CQR 3D7 strain was insignificant. PhcA also maintained its activity towards the D10 strain. TpcC exhibited about a three-fold increase in activity towards the CQI D10 strain (*p* = 0.0156) and had an inverse trend to that of TrcA when their activity towards 3D7 and D10 strains is compared. All three peptides, however, exhibited significant losses in activity towards the CQR Dd2 strain with TrcA >> PhcA ≥ TpcC in terms of resistance indices (Table 1). The selectivity index when toxicity towards COS cells are considered also followed that same trend (Table 1). It must be noted that the resistance indices between the D10 and Dd2 strain for QC and the three tyrothricin peptides showed a remarkable correlation ranging from 9–16 (Table 1). TrcA lost its activity towards the Dd2 strain to a point that at the concentration it inhibited the Dd2 strain one would expect selective hemolytic activity, similar to that of GS. The activity of GS against the three strains was similar (*p* > 0.5). GS was previously found to exhibit selective hemolytic activity towards infected erythrocytes [36]. GS and the three tyrothricin derived decapeptides also exhibited similar toxicity towards COS cells and human erythrocytes (Table 1).

### 2.2. Evaluation of Antimalarial Activity of the Cyclodecapeptides in Combination with Chloroquine

The IC_50_ results toward the three strains indicated that there is possibly at least one common mechanism of action/resistance between some of the Trcs and chloroquine. We carried out similar 48 h dose response assays using different combinations of TrcA, PhcA and TpcC with CQ to evaluate for interaction between these compounds (Table 2). At higher CQ concentrations there were antagonism between QC and the three tyrothricin-derived peptides with the fractional inhibition concentration (FIC) indices ibetween 1.3 and 3.5 (Table 2). TpcC exhibited the least antagonistic interaction with CQ and at the lowest CQ ratio it showed additive activity. The two Phe-Phe containing peptides, TrcA and PhcA, exhibited overt antagonism towards CQ (FIC index > 3, convex isobologram) at 1:1 and 1:2/1:5 CQ:peptide ratio, respectively (Table 2). At the lowest QC ratio TrcA and CQ still exhibited slight antagonism, but interestingly PhcA and CQ exhibited synergistic action with a FIC index of 0.54 (Table 2).

### 2.3. Microscopic Visualization of the Effect of Tryptocidine C on Parasite Morphology

TrpC exhibited the most robust activity profile towards the three *P. falciparum* strains with lowest resistance indices, although its activity was still slightly antagonized at the higher CQ concentrations. This larger and more polar cyclodecapeptide from the tyrothricin complex was more active than the two smaller Phe-rich peptides, TrcA and PhcA, and did not follow the SAR proposed by Rautenbach et al. [36]. These novel findings warranted further investigation into the mode of action and target(s) of this Trp-rich cyclodecapeptide.

Following the determination of the parasitemia through Giemsa staining and light microscopy, we observed that there was a delay in the occurrence of the schizont stage for the TpcC-treated cultures. Whereas in the untreated cultures (Figure 2A) the schizonts were observed from 12 h post incubation, they were only observed, albeit it al low concentrations, after 24 h in TpcC treated cultures (Figure 2B). However, it is possible that schizonts only occurred after 12 and before 24 h in the treated cultures. This could be because ring stage parasites appeared at 24 h post incubation for all cultures and ring stage progression was therefore unaffected, although it was at lower parasitemia for the treated cultures compared to the untreated ones (Figure 2, compare A and B at 24 h). This would have affected the turnover into trophozoites at the end of the 48 h incubation resulting in the pronounced difference in parasitemia observed between treated cultures and untreated cultures. Rautenbach et al. [36] observed a large difference in parasitemia between cultures treated with either TrcA or TrcC compared to untreated cultures after 21 h of incubation. In their work, they also observed that the TrcA and TrcC did not affect progression from starter ring cultures to trophozoite stage although the morphology of the resulting trophozoites was abnormal. In accordance with this, in the present study the parasites progressed from rings to trophozoite after 24 h but the parasitemia gradually declined suggesting a slow cytocidal rather than a static effect of the peptides on the parasites, correlating with previous studies [36].

Super-resolution structured illumination fluorescence microscopy (SR-SIM) was used to further study the influence on the malaria parasites, in particular to probe the TpcC influence on the CQ target, namely hemozoin in the parasite vacuole and putative peptide targets. These possible TpcC targets include the parasite membranes that were visualized with trypan blue as membrane impermeable dye (red fluorescence), neutral lipids and lipid bodies that are important for hemozoin crystallization using a neutral lipid binding LipidTOX dye (green fluorescence) and chromatin using SYTO9 Green membrane permeable nuclear stain (green fluorescence) as targets.

We confirmed previous studies [36] that the none of the parasite membranes or erythrocyte membrane was permeabilized or lysed by TpcC at a lethal concentration of 200 nM. There was no trypan blue leakage into the erythrocyte or parasite (Figure 3 and Figure 4) and the nuclear membrane of the parasite was not damaged as confirmed by the confinement of the parasites’ nuclear material within the nucleus as indicated by bis-benzimide (results not shown) and SYTO9 Green staining (Figure 3). These results indicated that TpcC, similar to TrcA [36], also acts by a slow non-lytic mechanism of action which inhibits maturation of the intraerythrocytic stages of *P. falciparum*.

More detailed analysis of the SR-SIM images revealed more information on the TpcC action and possible targets. In some of the images taken for the treated samples, we observed abnormality in the packing or compactness of the chromatin as indicated by the SYTO9 nuclear staining (Figure 3). This could suggest that the change in DNA packing is the consequence of the TpcC activity. For *P. falciparum* there was a deformation of the nuclear distribution with change in the compact nature of the nuclear material leading to a mixture of intense and diffuse staining of the chromatin (Figure 3E–L). In addition, we also observed that, unlike the discrete segregated distribution of chromatin seen in the untreated schizont (Figure 3D), the treated parasites exhibited a loss of the uniform segregation and spherical morphology of the schizont chromatin (Figure 3J–L). This could indicate that some of the merozoites derived from this schizont will not be viable and explain the gradual decrease in parasitemia that was observed (Figure 1B). A similar effect on schizont chromatin morphology has been observed with GS and TrcA treatment in previous studies [36]. Apart from overt changes in parasite morphology, we also observed some abnormality in the shape and compactness of a dark structure assumed to be the hemozoin crystal in the treated samples (white arrows in Figure 3F,H). Unlike in the untreated sample, in some instances only observed in the treated samples there were dense dark dispersed and irregular elongated structures within the SYTO9 nuclear stain (Figure 3F,H) rather than more confined dark structures normally observed in the intra-erythrocyte trophozoites [39]. This suggested a compromise in the parasite’s ability to form a single hemozoin crystal and this could favor heme toxicity following the parasite’s digestion of hemoglobin [40]. Chloroquine has also been observed to cause the clumping of smaller hemozoin crystals [41,42,43].

Due to the possibility that one of the molecular targets of the Trcs are located in the food vacuole, as suggested by increased Trc resistance by CQ resistant *P. falciparum* and the CQ-Trc antagonism, it is possible that part of non-lytic mode of action of the Trcs involves heme toxicity, similar to that of chloroquine. Competition for this target may lead to some of the observed antagonism. It has been suggested that hemozoin crystals in the malaria parasite are associated with neutral lipid droplet-like structures [44] within the food vacuole and that these lipid droplets along with phospholipid membranes are involved in the process of heme crystallization both in vivo and in vitro [44,45,46,47,48,49,50].

To have a closer look at the parasite lipid structures we used a SR-SIM method to visualize the effect of TpcC on the neutral lipid accumulation using the green fluorescent LipidTOX neutral lipid stain. A striking increase in phospholipid and neutral lipid content has been reported as one of the main changes that occur following malaria infection of human erythrocytes [51,52,53,54]. This increase in lipid content is geared towards synthesizing the complex membranous system of the *Plasmodium*-infected erythrocyte [51]. Our results agree with previous observations that neutral lipids are not detectable in normal erythrocytes [51,52,54], as we only found the stained lipids within infected erythrocytes (Figure 4). These detected neutral lipids include fatty acids, diacylglycerol and triacylglycerol, with the former two closely associated with the food vacuole originating from the digestion of phospholipids from transport vesicles used for hemoglobin ingestion [50,51,52,53,54]. We observed that most of the stained neutral lipids accumulated in the intra-erythrocytic cellular space formed part of the parasites’ membranes (Figure 4A).

This malarial parasite morphological details observed were unlike any yet described, to the best of our knowledge. Notably, regular and circular shapes of the lipid structures were observed in normal untreated parasites (Figure 4A–E), with trophozoites having multiple circular structures (Figure 4B,C), sometimes associated with an intensely stained lipid droplets or bodies. The segregated pre-merozoite structures in the schizont shown in Figure 4E were clearly outlined by the neutral lipid stain. After treatment with TpcC the lipid structures were distorted and diffused (Figure 4F–H). There were also multiple intensely stained lipid droplets/bodies visible (Figure 4G,H,J), while some parasites did not show any straining (Figure 4F,I).

## 3. Discussion

The three cyclodecapeptides from tyrothricin had different activities towards the three different *P. falciparum* strains. Differences in the infected erythrocytic membrane must be considered as the cyclodecapeptides must first recognize and cross this membrane to exerts its action on the intraerythrocytic parasite. GS selectively acts on the infected erythrocytic membrane [36], and as expected, had a resistance index of near one and statistically similar IC_50_ values against the three strains (Table 2). One can therefore assume that the three strains lead to a similar lipid composition for the infected erythrocyte for recognition by GS and for initial interaction with the three cyclodecapeptides. This therefore cannot explain the loss in activity of these cyclopeptides peptides. However, if a primary target, other than the transport of the Trcs over the erythrocyte membrane, is changed or the interaction with the peptides with the intracellular target is limited by either target concentration or a resistance mechanism, this probably would lead to a major loss in activity.

Alternatively, neutral lipid structures could also be targeted due to the lipophilicity of the Trcs and analogues. The formation of the neutral lipid structures depends on the stage of the parasite and peak during the mid- and late-trophozoite stages [53,54] which could explain the stage selective activity of the Trcs which is more active towards trophozoites and schizont stages [36]. The structures observed in this study with the LipidTOX dye seemed to extend from the parasite to the erythrocyte membrane and may not be limited to neutral lipid bodies within the food vacuole. The parasite is also known to induce the formation of an interconnected network of turbovesicular membranes within 33 h post-infection which indeed runs from the parasites vacuolar membrane to the erythrocyte membrane and are involved with transport of nutrients to the parasite such as nucleosides and amino acids [43]. All these vital roles played by the parasite membranes, including neutral lipid membranous structures could be interfered with. This would result of in the change in cell morphology that was observed following treatment with the membrane active TrcC and TrcA in a previous study [36] and TpcC in this study.

The fact that we observed resistance in the erythrocytic stage of *P. falciparum* towards the three tyrothricin derived cyclodecapeptides is concerning, especially if membranes/lipids are the target. A major change in lipid composition would be necessary for this type of target-dependent change and resistance. However, this resistance seems to depend on peptide sequence, size and amphipathic character, with resistance indexes the highest for TrcA followed by PhcA and TpcC. Substituting smaller and non-polar L-Phe^3^-D-Phe^4^ dipeptide unit in the tyrothricin cyclodecapeptides with the larger and more polar L-Trp^3^-D-Trp^4^ dipeptide unit with a greater hydrogen-bonding ability at position 4 resulted in a change in activity and modulated *P. falciparum* resistance. The change in TpcC activity depended on the strain with the CQI D10 strain being more sensitive to TpcC than to the other peptides. The Phe-rich TrcA activity mirrored that of chloroquine with IC_50_ sequence CQS 3D7 ≤ CQI D10 << CQR Dd2. The substitution of L-Tyr^7^ with L-Phe^7^ led to lower resistance and indicates that the 3D7 activity could be determined by a specific target interaction depending on L-Phe^3^-D-Phe^4^ and L-Tyr^7^, while L-Phe^7^ is important for activity against D10, helping to maintain the activity. This sequence specificity could explain the lower activity of TpcC that contains L-Trp^3^-D-Trp^4^ and L-Trp^7^ towards the 3D7 strain. Rautenbach et al. [36] suggested that such a putative target relies on the D-Phe^4^ of the aromatic dipeptide unit and Orn^9^ in the conserved pentapeptide. The loss of this interaction due to a change in the 3D7 target in the D10 and Dd2 strains could relate to the observed resistance to TrcA. The 3- and 48-fold loss in activity of TrcA towards the CQI D10 and CQR Dd2 strains, respectively, is consistent with the hypothesis that a specific target is lost. However, the Phe-rich PhcA, containing four Phe residues, was much less affected and this lipophilicity would favor a second target such as membrane interaction since Phe has been found to integrate more efficiently into membranes [55,56]. Peptides containing Phe will have a preference for the water-lipid interface [36,56,57]. A peptide like PhcA could be more active in preventing the hematin polymerization to form hemozoin by interaction with lipid droplets. However, a very tight membrane/lipid association does not always translate into better multi-target activity. There is need for an optimal amphipathicity which still allows for efficient membrane integration and translocation to an internal molecular target [58,59,60].

Membrane interaction or a sequence specific target does not explain the activity of TpcC with L-Trp^3^-D-Trp^4^ and Trp^7^ in its structure. TpcC was less active against the CQS 3D7 strain following the SAR proposed by Rautenbach et al. [36] and this may be because of weaker target interaction, as discussed above. However, it exhibited the best activity against the CQI D10 and CQR Dd2 strains. The SR-SIM analysis of TpcC action indicate an influence on chromatin morphology. TpcC has three Trp residues and it is known that Trp can interact with DNA [61]. Wenzel et al. [62] also found that TrcC, with two Trp residues, led to condensation of the *Bacillus subtilis* nucleoid, indicating that these peptides could interact with DNA. It is therefore possible that the better interaction of this Trp-rich peptide with the parasite DNA could lead to better activity towards the D10 and Dd2 strains. From the SR-SIM analysis of TpcC action on the D10 stain, it could be seen that the chromatin packing was indeed disturbed. However, many of the parasite’s other structures were also influenced, including the parasite vacuole, without leading to membrane permeabilization (Figure 3 and Figure 4). In particular, the neural lipid structures and distribution were highly influenced and the shizonts lacked the typical knobbed rosette morphology. Rautenbach et al. [36] did not report such overt shizont morphology changes by 3D7 strain exposure to TrcA and TrcC. Our new observation indicates a complex, possibly multi-targeted intracellular mode of action. In another study, fluorescence microscopy of a labelled cationic peptide, ΔFd, show that this peptide crossed various membranes, including the parasitophorous vacuolar membrane, the parasite’s plasma and nuclear membranes to interact with DNA of the parasite [23]. Similar to our observations, schizonts following treatment with ΔFd contained knobs of amplified DNA lacking the typical symmetric rosette appearance which could not proliferate into merozoites for release and re-invasion [23]. These results and previous results from our group [36] suggested that the Trcs and TpcC could act in a similar manner.

An intracellular mode of action relies on transport to the target and compromised transport could have led to loss in the cyclodecapeptide activity against the more CQ resistant *P. falciparum* strains. It is known that some of the compounds that rely on the drug/metabolite transporter family, of which *Pf*CRT and *Pf*MDR1 are members, are amino acids, weak bases and positively charged organic ions [63], as well as small peptides which are produced following degradation of hemoglobin in the food vacuole [64].

*Pf*CRT is implicated in the resistance of *P. falciparum* to several antimalarial drugs such as quinine and quinidine [65], amodiaquine [66], halofantrine and mefloquine [67]. The main mutation that brings about CQ resistance is a Lys^76^ to Thr^76^ mutation in the *Pf*CRT which is suggested to lead to a loss of a positive charge in a putative pore-forming transmembrane domain (in the food vacuole membrane) facilitating the escape of diprotonated CQ from the parasite’s food vacuole [4,63]. Mutated *Pf*CRT has been shown to directly transport the radio-labelled YPWF–NH_2_, a peptide rich in aromatic amino acids [68], not unlike the peptides in this study. Conversely, VDPVNF (VL-6, a hemoglobin fragment) was also observed to *cis*-inhibit CQ transport through *Pf*CRT [69]. If the food vacuole is one of the cyclodecapeptide targets, it could be that size matters with the smaller PhcA and TrcA, both containing a fPFf sequence being transported out or escaping out of the food vacuole by mutated *Pf*CRTs in the Dd2 strain, with the larger TpcC with fPWw in its sequence being less affected.

Another transporter protein found in the food vacuole membrane and known to contribute to CQ resistance is *Pf*MDR1 (*Plasmodium falciparum* multi-drug resistance transporter 1). *Pf*MDR1 also known as Pgh-1 is the P-glycoprotein [70,71] that is implicated in CQ resistant phenotypes [64,66] such as the CQR strain Dd2, a clone obtained from the CQR W2 strain first isolated from Southeast Asia [72,73]. An Asn to Tyr substitution at position 86 in the *Pf*MDR1 has been suggested to contribute to increased CQ resistance in the Dd2 parasite. This was observed to completely change the substrate affinity of *Pf*MDR1 from a quinine and CQ transporting ability to a halofantrine transporting function following expression of the gene in *Xenopus laevis* oocytes [74]. This transporter is also implicated in artemisinin resistance [8]. The role of *Pf*MDR1 in CQ resistance has been shown to involve diminished uptake of the drug into food vacuole and active transport out due to mutations in this transporter [64,75]. There is evidence that *Pf*MDR1 functions in transport of small peptides which arise from incomplete catabolism of hemoglobin from the food vacuole to the cytoplasm [76,77]. The presence of an analogous MDR1 in mammalian cells led to cross-resistance to a synthetic tripeptide (N-acetyl-leucyl-leucyl-norleucinal) [78], gramicidin D (15 residue linear peptide from tyrothricin) [79,80] and valinomycin (a cyclic dodecadepsipeptide) [79]. Our results suggest that the Trcs may be among the substrates for the CQ transporters and may have an intracellular molecular target in the digestive vacuole and other lipid structures. Mutations in these transporters could lead to resistance. It is possible that the aromatic residues at position 3, 4 and 7 is relevant to the process of transport and target interaction in the cytoplasm and food vacuole. The larger size and lower lipophilicity of TpcC compared to PhcA and TrcA makes it less susceptible to mutations linked with CQ resistance in the D10 and Dd2 strains.

With the assumption that the cyclodecapeptides have an intracellular target, that may involve the food vacuole, neutral lipids and hemozoin polymerization (hematin crystallization), the CQ antagonism is not unexpected. It has been suggested that the mode of antimalarial action of CQ and other quinolines involves entry into the neutral lipid microenvironment which interferes with the hematin crystallization process that is facilitated by a complex neutral lipid structure [44]. Decreased access of CQ to its target, hematin which is produced in the food vacuole following hemoglobin digestion [81,82] and/or change of proton flux or CQ influx at the parasite’s food vacuole membrane [83,84] has been also linked to QC resistance. Therefore, another way in which these peptides could antagonize CQ action is by interfering with the food vacuole membrane integrity and leading to leakage of CQ from the food vacuole; thereby limiting the accumulation of CQ that is essential to its antimalarial activity. This could explain the observed antagonism between the Trcs and CQ if they compete for interaction with the neutral lipid structures. Otherwise, the Trcs could interfere with hemoglobin ingestion and/or hematin crystallization by disturbing the network of membranes made of neutral lipids. Trp is less lipophilic than Phe and have shallower interaction with membranes [55,56]. This could benefit non-lytic TpcC activity such as inhibiting hematin crystallization at the lipid interphase, but not disrupt the membrane/lipid interphases as much to cause a large QC loss of activity, leading to antagonism, as found for more hydrophobic TrcA and PhcA. Alternatively, the Trcs could lead to increased permeability of the food vacuole allowing CQ escape. As synergism was limited to a low CQ:PhcA ratio, it is ruled out that these peptides play a major role in obstruction of *Pf*CRT-mediated CQ transport. Previously it was found that the radio-labelled linear tetrapeptide YPWF-amide (endomorphin analogue) competed for CQ transport followed by WHWLQL (α1-mating factor peptide) observed for the transporter protein functionally expressed in oocytes of *Xenopus laevis* [68]. All three peptides contained aromatic amino acids with YPWF-amide sharing some sequence identity with fPFf and fPWw in TrcA/PhcA and TpcC, respectively, so these peptides could still be substrates of *Pf*CRT leading to resistance. These results support the hypothesis that the Trcs and CQ may share at least one of their targets, probably in the parasite food vacuole, and when both compounds are present in this target organelle there is an antagonistic effect.

## 4. Materials and Methods

### 4.1. Materials

Tyrothricin (extracted from *Brevibacillus parabrevis*), gramicidin S (from *Brevibacillus brevis* (Nagano)), and Corning Incorporated^®^ cell culture cluster non-pyrogenic polypropylene microtiter plates, bis-benzamide trihydrochloride (Hoechst stain) and trifluoroacetic acid (TFA, >98%) were obtained from Sigma (St. Louis, MA, USA). All the chemicals used to prepare the RPMI-1640 culture media (RPMI 1640 medium, glucose, HEPES, albumax II, hypoxanthine, NaOH, gentamycin, and sodium bicarbonate), sodium lactate, potassium chloride, NaCl, L-lactic acid, nitro blue tetrazolium (NBT), phenazine ethosulfate (PES), 3-acetylpyridine adenine dinucleotide (APAD), D-sorbitol, Dulbecco’s modified Eagle’s Medium (DMEM), 0.4% trypan blue solution, and DNA interchelator Giemsa stain mixture were obtained from Sigma-Aldrich (St. Louis, MA, USA). The synthetic tyrocidines were supplied by GL Biochem (Shanghai) Ltd., China. Resazurin reagent (CellTiter Blue™) was from Promega (Madison, WI, USA). Sterile red standard cap 250 mL Cellstar tissue culture flasks, sterile Cryo.s PP tubes and sodium hydrogen phosphate were from Greiner Bio-One GmbH, Germany. Glycerol (AnalaR grade) was obtained from BDH Chemicals Ltd. Triton X-100 came from BDH Laboratory Supplies, Poole, England. Tris-HCl buffer was obtained from Boehringer Mannheim or Roche. Acetonitrile (ACN) (HPLC-grade, far UV cut-off) came from Romill Ltd. (Cambridge, UK). To obtain analytical grade water, water was filtered from a reverse osmosis plant via a Millipore Milli-Q water purification system (Milford, CT, USA). Ethanol (>99.8%) was supplied by Merck (Darmstadt, Germany). Culture dishes and 0.2 μm–25 mm sterile cellulose acetate membrane syringe filters were obtained from Lasec (Cape Town, South Africa) and microtiter plates (NuncTM-Immuno Maxisorp) were from AEC Amersham (Johannesburg, South Africa). Falcon^®^ tubes were from Becton Dickson Labware (Lincoln Park, IL, USA). Fetal calf serum and penicillin–streptomycin were from Gibco BRL (Gaithersburg, MD, USA). Sterile VacuCap^®^ 90PF filter unit w/0.8/0.2 µm Supor^®^ membrane was obtained from Pall Corporation (Pall Europe Ltd., Portsmouth, UK). SYTO^®^ 9 green fluorescent nucleic acid and HCS LipidTOX™ neutral lipid stains were obtained from Invitrogen (Carlsbad, CA, USA). Whole A^+^ blood stored in anticoagulant (citrate phosphate dextrose) containing enriched erythrocyte fraction in saline adenine-glucose-mannitol red blood cell preservation solution was donated by the Western Cape Blood services (or National Health Laboratory Services in South Africa). Blood used for all sets of experiments in this study was expired samples of fully tested anonymous human blood donations complying with all relevant legislation. At the time of this study the use did not constitute a breach of ethics and no ethical approval was required. Asexual erythrocytic stage chloroquine sensitive (CQS) *Plasmodium falciparum* D10 and 3D7 as well as chloroquine resistant (CQR) *P. falciparum* Dd2 (Asian/African) were kindly supplied by Prof. Peter Smith from the Division of Pharmacology, University of Cape Town. COS-1 cells were provided by Prof. Pieter Swart from the Department of Biochemistry, University of Stellenbosch.

### 4.2. Methods

#### 4.2.1. Parasite Culturing

Culture media preparation: The composition of the media used to culture *P. falciparum* was RPMI-1640 (10.4 g/L) supplemented with glucose (4 g/L), HEPES (6 g/L), albumax II (5 g/L), hypoxanthine (0.4 g/L, dissolved previously in 1 mL of 1 N NaOH), gentamycin (50 mg/L), and sodium bicarbonate (2.1 g/L). The media was made up in analytical quality water; the pH was adjusted to 7.2–7.3 and sterility was achieved by filtering through a 0.2 µM filter [85,86].

Preparation of blood: Anonymous A^+^ donor blood (300 mL enriched erythrocyte fraction containing 63.0 mL citrate phosphate dextrose anticoagulant and 100 mL saline-adenine-glucose-mannitol red blood cell preservation solution) from the Western Cape Blood services (or National Health Laboratory Services in South Africa) conforming to relevant legislation and ethics were utilized during all experiments of this study. Routinely A^+^ erythrocyte enriched blood of not older than two weeks was used since the *P. falciparum* cultures did not develop well on older erythrocytes [87]. Prior to use, the blood was washed twice in parasite culture media by centrifugation at 1300× *g* for 5 min per wash followed by decantation of supernatant (containing plasma and buffy coats if present).

*P. falciparum* culturing procedure: Culturing was carried out using normal sterile techniques according the methods of Trager and Jensen [86] and Lambros and Vanderberg [88]. Freezer stock cultures of CQS (D10 and 3D7) and CQR (Dd2) *P. falciparum* were thawed in a water bath at 37 °C and the cultures were transferred to 50 mL falcon tubes. The osmotic potential of the thawed parasite freezer stocks in glycerol was progressively reduced by stepwise dilution in three saline solutions prepared in analytical quality water namely solution A (12% NaCl), solution B (1.8% NaCl), and solution C (0.9% NaCl and 0.2% glucose) [89]. Five volumes of solution A was first added drop wise with swirling to 1 volume of thawed cultures and allowed to stand for 5 min followed by slowly adding 10 mL of solution B. Subsequently the combination was centrifuged at 400× *g* for 5 min followed by aspiration of the supernatant. Ten mL of solution C was gradually added to the residual pellet with moderate swirling. The mixture was centrifuged at 400× *g* for 5 min followed by aspiration of the supernatant. The residual culture was washed with 20 mL of culture media by centrifuging at 400× *g* for 5 min and then returned to culture media for culturing.

Culturing involved incubating the parasitized blood (at 3–4% hematocrit, i.e., 3–4 mL blood/100 mL media) in sterile red standard cap 250 mL Cellstar tissue culture flasks at 37 °C under a gas mixture of 3% O_2_, 4% CO_2_, and 93% N_2_ without shaking. The initial culture from freezer stocks was left to stand for 3–4 days. Culturing was carried out continuously for not longer than three months from one stock culture to avoid genetic modification 51. The parasite development and parasitemia (number of parasites in the infected erythrocytes expressed as a percentage of the normal erythrocytes) were followed by taking samples of the cultures to prepare thin blood smears on microscope slides for staining with Giemsa and viewing under oil immersion at the 100× objective lens of a light microscope according to the methods of Reilly et al. [90]. From 0.5% parasitemia, the media was replaced daily by centrifugation at 750× *g* for 3 min, aspiration of the supernatant and addition of fresh media.

It was important to maintain the cultures at the same developmental asexual stage and this was achieved by addition of 5 volumes of 5% D-sorbitol to 1 volume of pelleted parasite infected erythrocytes. In vitro synchronization was carried out only when most of the parasites were at the ring stage (because the membranes of erythrocytes containing ring stage parasites are less permeable to solutes [91] according to the methods of Lambros and Vanderberg [88]. Following addition of sterile 5% D-sorbitol to the pelleted erythrocytes, the mixture was gently swirled and allowed to stand for 5 min in a water bath at 37 °C. Next, the mixture was centrifuged at 750× *g* for 3 min, the supernatant was aspirated, and the pellet was washed once with culture media by centrifugation at 750× *g* for 3 min and aspiration of supernatant. The pellet was then returned to culture media and maintained according to the culturing procedures mentioned above. Synchronization was limited to once a week due to the finding by Makowa [87] that D-sorbitol favors the development of resistance to chloroquine.

Once a high parasitemia (>10%) was achieved, the cultures were either frozen away in glycerol at −80 °C for culture preservation or used for dose response assays. Freezing involved ring stage parasites only as their membranes are more robust according to the method of Diggs et al. 53. The glycerolyte medium in which the parasites were stored at 80 °C was made of 1.6% sodium lactate, 0.03% KCl, 1.38% sodium dihydrogen phosphate, 57% glycerol adjusted to pH of 6.8. The solution was sterile filtered through 0.22 µm filters. One volume of glycerolyte medium was added to 3 volumes of pelleted infected erythrocytes in a drop wise manner with constant swirling. The mixture was allowed to stand for 5 min followed by addition of 2 volumes of glycerolyte medium. The cultures were stored in Cryo.s PP tubes by storage at 80 °C until required for use.

#### 4.2.2. Peptide Preparation

The TrcA from commercial tyrothricin were purified and characterized as previously according to the method of Rautenbach et al. [36]. The other natural analogues (TpcC, TrcA and Phc A) were extracted from cultures of *Brevibacillus parabrevis* ATCC 8185 under specific nitrogen supplementation conditions, purified and analyzed as described by Leussa and Rautenbach [92]. The same preparations of TrcA, PhcA and TpcC, as prepared by Leussa and Rautenbach [92], were used in this study.

The analytically weighed purified peptides were subsequently used to analytically prepare stock solutions of 2.00 mM (purified peptides) or 2.00 mg/mL (Trc mixture or Trc mix) with 40% *v/v* ethanol in analytical quality water (Trcs) or with analytical quality water (gramicidin S). Subsequently the stock solutions were used to construct quadrupling dilution series in polypropylene 96 multi-well plates containing the supplemented RPMI-1640 culture medium used for culturing the *P. falciparum*, but lacking albumax II. Preparation of stock solutions and subsequent dilution was performed about 30 min prior to each assay.

#### 4.2.3. Determination of Antimalarial and Hemolytic Activities of Peptides

The antimalarial assays were carried out according to the method of Nkhoma et al. [93]. Once the cultures reached a parasitemia of 5–15%, they were synchronized four days prior to the day of the assay as described above which allowed the cultures to recover from the stress caused by the D-sorbitol. On the day of the assay, the parasitemia was determined following observation by light microscopy after Giemsa staining of a thin blood smear from the culture. The cultures at young trophozoite stage were diluted to 2% parasitemia and 2% hematocrit by addition of fresh RPMI medium and uninfected erythrocytes. This culture suspension was distributed in 96-well culture plates at 90 µL/well. The peptides and CQ diluted in culture medium lacking albumax II were added to the cultures in triplicate at 10 µL/well to give a total volume of 100 µL/well. Peptide solvent and medium (10 µL/well) was used as sterility control and in growth control and 10 μL 1.0 mM GS was used as positive control. Assays were carried out in at least three biological repeats each consisting of three technical repeats to ascertain reproducibility.

For the activity determination against parasites, plates were incubated at 37 °C for 48 h under an atmosphere of 3% CO_2_, 4% O_2_ and 93% N_2_. The culture suspension was frozen away at −20 °C until determination of the residual lactate dehydrogenase activity of the parasites by the Malstat assay, as adapted from Gomez et al. [94].

For the Malstat assay, the plates were transferred to −80 °C for 1 h following overnight storage at −20 °C to ensure that all the suspensions in each well of the 96-well plate were frozen. The plates were then thawed at room temperature. The Malstat reagent consisted of 200 μL Triton X-100, 2 g L-lactic acid (substrate), 0.66 g Tris-HCl buffer and 0.011 g of 3-acetylpyridine adenine dinucleotide (APAD) (coenzyme for parasite lactate dehydrogenase) in 100 mL analytical quality water set to a pH of 9.0 [93]. The lactate dehydrogenase reaction was initiated by addition of a second solution, NBT/PES solution composed of 1.96 mM nitro blue tetrazolium (NBT) and 0.24 mM phenazine ethosulfate (PES) which was always protected from light. A 15 µL aliquot of the thawed and properly mixed suspension from the assay plates was added to 100 µL of the Malstat reagent, followed by addition of 25 µL of the NBT/PES solution. The reaction mixtures were then incubated at room temperature in the dark for 30 min followed by spectrophotometric measurement of the absorbance of the reduced APAD at 620 nm using a Model 680 Microplate reader from BioRad.

To determine the hemolytic activity of the peptides, the duplicate 96-well plates with samples added to 90 μL 2% hematocrit per well were incubated for one hour at 37 °C and centrifuged using a swing-out rotor at 200× *g* for 3 min. The supernatant was diluted 1:8 in analytical quality water in separate 96-well plates and after mixing, the absorbance of the suspension was determined at 405 nm.

#### 4.2.4. Determination of Toxicity

COS-1 cells were cultured in DMEM containing 0.9 g/L glucose, 0.12% NaHCO_3_, 10% fetal calf serum and 1% penicillin–streptomycin at 37 °C and under an atmosphere of 5% CO_2_ [95]. Twenty hours before the assay, the cells were transferred to 96-well plates at a density of 1.5 × 10^4^ cells/well. The peptides from the dilution series were added to each well and incubated for 24 h. The cell viability was measured by adding 20 μL/well resazurin reagent (CellTiter Blue™), followed by four hours incubation at room temperature in the dark and measurement of the absorbance at 560 nm and 600 nm in the BioRad microplate reader [36].

#### 4.2.5. Assessment of Dose–Response Data

The data obtained from spectrophotometric measurement of the plates following the lactate dehydrogenase (Malstat) assay were converted to percentage growth inhibition using the data analysis method described by Rautenbach et al. [96]. The background was obtained from control wells in which the parasites were killed with 100 µM GS while total growth was from the wells that were not submitted to peptides. The background for the hemolysis evaluation was from wells in which the uninfected erythrocytes were lysed using 100 µM GS while growth control was from the wells with uninfected erythrocytes that received no peptides [36]. The percentage COS-1 cell death (CellTiter Blue™ assay) was similarly calculated from absorbance values. All the dose–response assay data were evaluated using GraphPad Prism 4.03 (GraphPad Software, San Diego, CA, USA) followed by non-linear regression and sigmoidal curves were fitted (having variable slope and a constant difference of 100 between the top and bottom plateau). Following calculation of the 50% *P. falciparum* inhibitory concentration (IC_50_), 50% hemolytic concentration (HC_50_) and 50% lethal concentration (LC_50_) for COS-1 cells [96].

#### 4.2.6. Interaction between Chloroquine and Selected Tyrocidine Analogues

In order to determine the nature of Trc-CQ interaction, IC_50_ values were derived from dose–response curves of fixed ratios of Trc and CQ for fractional inhibition concentration (FIC) determination according to an adaptation of the methods by Chawira and Warhust [97] and Fivelman et al. [98].

CQ at 4000 ng/mL (2.0 µM) and 500.0 μM peptide were used to prepare doubling and quadrupling dilution ranges, respectively in supplemented RPMI-1640 culture medium used for culturing the *P. falciparum*, but lacking albumax II. These were then mixed in an chequerboard fashion to determine the 1:0; 0:1 1;1, 1:2; 1:5 and 1:10 ratios of CQ:peptide for combined IC_50_s. The CQ, peptide or drug:peptide combination (10 μL) was added to 90 μL of the cell suspensions, incubated for 48 h at 37 °C under an atmosphere of 3% CO_2_, 4% O_2_ and 93% N_2_ and analyzed for growth inhibition as previously described using the Malstat assay. The IC_50_ values and standard error of the mean for the various CQ-peptide combinations were determined from the dose–response curves plotted and analyzed using GraphPad Prism^®^ 4.03 (GraphPad Software, San Diego, CA, USA). A minimum of three technical repeats and three biological repeats were carried out for each combination experiment.

Two fractional inhibition concentration (FIC) values were calculated for each of the five CQ-peptide combination ratios, one for CQ and the other for the peptide according to Equations (1) and (2) below from Bell [99]:FIC_CQ_ = IC_50_([CQ] in combination)/IC_50_CQ ([CQ] alone)(1)
FIC_Peptide_ = IC_50_ ([Peptide] in combination)/IC_50_Peptide ([Peptide] alone)(2)

The FIC values were used to compute the FIC index which is the sum of FICs of CQ and each peptide using Equation (3) below [98,100]:FIC index = FIC_CQ_ + FIC_peptide_(3)

The magnitude of the FICI determined the nature of the CQ-peptide interaction as being either synergistic (FICI < 1.0), antagonistic (FICI > 1.0) or additive (FICI = 1.0) 64–66. However, a more conservative interpretation requires that FICI ≤ 0.5 indicates absolute synergy, 1 > FICI > 0.5 shows slight synergy, FICI = 1 means additive activity, 1< FICI < 4 is interpreted as non-interactive to slight/moderate antagonism, while FICI ≥ 4 indicates absolute antagonism [87,99,100,101]. The shape of the isobolograms also provided an indication of the nature of the interactive effect of CQ and the peptides with a concave curve for synergy, a linear line for an additive to non-interactive effect or convex curve for antagonism with deviation of the curves from the additivity line indicating the strength of the interactive effect [99].

#### 4.2.7. Evaluation of TpcC Activity Using Microscopy

To determine the effect of the most active Trc analogue (according to IC_50_ from the antiplasmodial assay) on cultured *P. falciparum* and to assess its mode of action, the synchronized CQI D10 strain cultures at trophozoite stage (2% parasitemia, 1% hematocrit) were incubated with the drug at non-hemolytic concentrations 2-fold above its IC_50_ and evaluated using both light and fluorescence microscopy according to methods described by Rautenbach et al. [36] and Wiehart et al. [26] with modifications as described below.

For light microscopy, following addition of TpcC, aliquots were collected from the cultures at different time-points for preparation of Giemsa-stained blood smears. The assay was carried out in duplicate and lactate dehydrogenase activity of the residual cultures was determined after 48 h.

Prior to staining for fluorescence microscopy, the cultures were centrifuged at 300× *g* for 4 min, then the supernatant was discarded and cells were suspended to the same percentage hematocrit in a solution made of one part analytical quality water and three parts sterile PBS (pH 7.2–7.3) in order to reduce solute concentration.

The fluorescence microscopy involved staining suspended cultures (TpcC treated and untreated) prepared as described above (except for not being synchronized within the last 48 h of culturing) with combinations of trypan blue (red fluorescence) and either SYTO^®^ 9 green-fluorescent nucleic acid stain or LipidTOX™ neutral lipid stain (green fluorescence). To view the neutral lipids, the cells were stained with LipidTOX™ (stock solution dilution at 1:200), incubated for 30 min at room temperature before staining with trypan blue at final concentration of 0.002% *v/v*. For nucleic acid staining, the cells were stained with SYTO^®^ 9 (stock solution dilution at 1:1000) and trypan blue at final concentration of 0.002% *v/v* and incubated at room temperature for 15 min. Super-resolution structured illumination (SR-SIM) fluorescence microscopy was carried out as follows: thin (0.1 mm) z-stacks of high-resolution image frames were accumulated in 5 rotations using an alpha Plan-Apochromat 100×/1.46 oil DIC M27 ELYRA objective, employing an ELYRA S.1 (Carl Zeiss Microimaging) microscope equipped with a 488 nm laser (100 mW), 561 nm laser (100 mW) and Andor EM-CCD camera (iXon DU 885). Images were re-enacted using ZEN software (black edition, 2011, version 7.04.287) based on a structured illumination algorithm [26]. Image projections and animations were carried out on reconstructed super-resolution images in ZEN.

## 5. Conclusions

In this study, we again found that TrcA and two analogues TpcC and PhcA showed potent antimalarial activity that was sequence specific and primarily non-lytic. The results confirm previous findings that natural Trc analogues had significantly higher antiplasmodial activity than GS [36]. In this study, we found the size and hydrophobicity dependent SAR for the Trcs, proposed by Rautenbach [36], did not hold true for *P. falciparum* strains other than 3D7. For activity against the 3D7 the prerequisite seems to be size, depending on the aromatic residues 3, 4 and 7. The Tyr^7^ in TrcA seem to be related with loss in activity correlating with CQ resistance and antagonism, indicating a shared target and/or resistance mechanism in which the phenolic group plays a role. Phe^7^ in PhcA correlated with maintenance of activity towards both 3D7 and D10 strains, but overt antagonism towards CQ, again indicating a shared target. Trp^7^ in TpcC correlated with improved peptide selectivity and activity towards more the more resistant strains, without overt antagonism towards CQ, but leading to similar morphology changes than the other Trcs. The disorganization of chromatin could account for halted growth in late trophozoite/early schizont stage and as disorganization of neutral lipid structures could account for change in the uniform appearance of the hemozoin crystal. The latter supports that the Trcs and CQ could have a common target in the malaria parasite. The differing activities of these cyclic peptides towards different *P. falciparum* strains could be due to a different emphasis on specific targets in a multiple mode of activities, with the activity depending on the aromatic residues in positions 3, 4 and 7 and its interplay with the parasite targets. This new knowledge on these natural cyclic could also benefit the design of unique small peptidomimetics in which activity and resistance can be modulated.

## Figures and Tables

**Figure 1 antibiotics-11-00801-f001:**
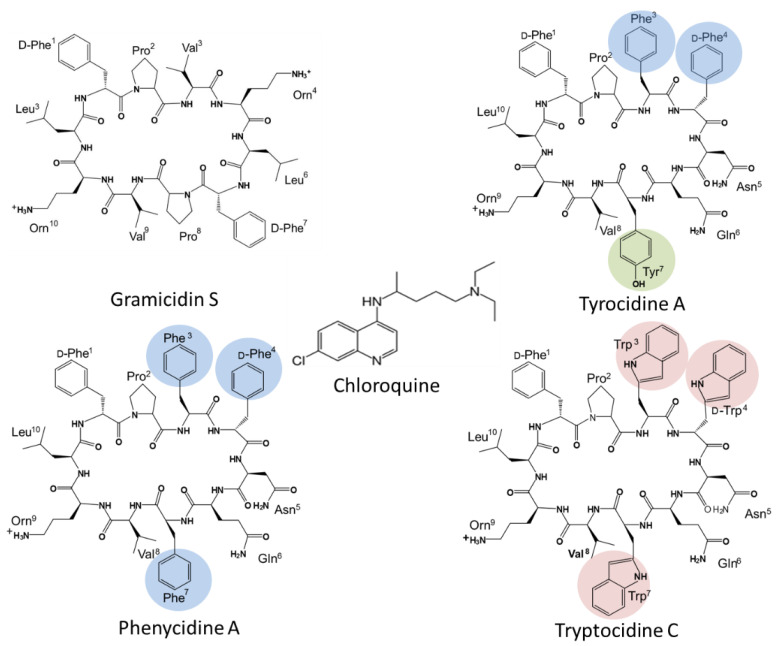
The chemical structures of gramicidin S (GS), tyrocidine A (TrcA), phenycidine A (PhcA), tryptocidine C (TpcC), and chloroquine (CQ).

**Figure 2 antibiotics-11-00801-f002:**
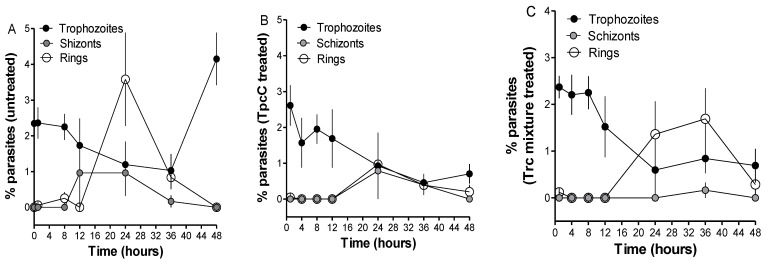
Distribution of trophozoite, schizont and ring parasite stages over time from 0 to 48 h following treatment of synchronized *P. falciparum* D10 cultures at trophozoite stage incubated without (control) (**A**), with 200 nM of TpcC (**B**) or 200 ng/mL Trc mixture (**C**). Each data point represents the mean ± SEM of parasite counts made within 8–13 regions on the microscope slide from two cultures of each treatment.

**Figure 3 antibiotics-11-00801-f003:**
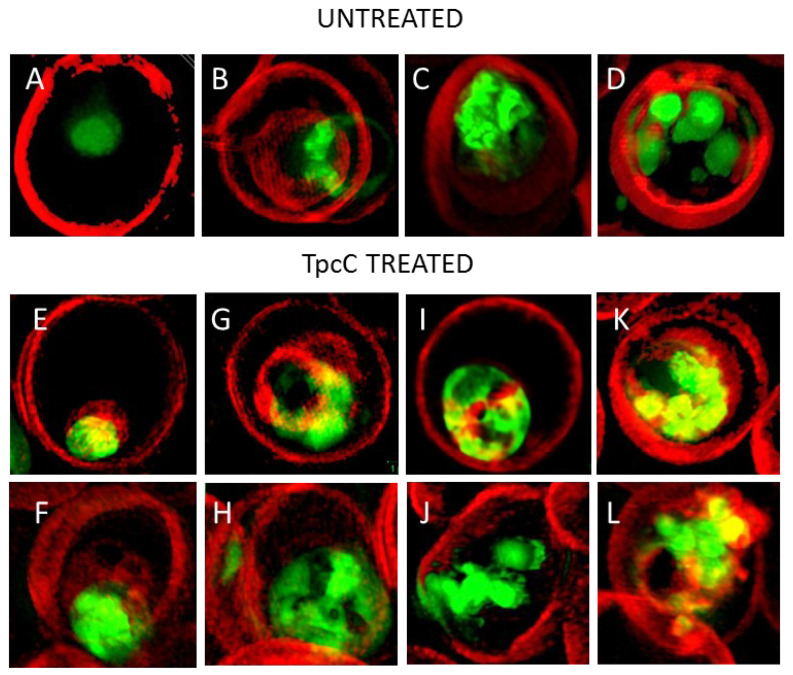
Super-resolution structured illumination fluorescence microscopy images of late intra-erythrocytic stages of *P. falciparum* D10 stained with the membrane impermeable fluorescent dye trypan blue (red) and the permeable nucleic acid fluorescent dye SYTO9 (green). Images (**A**–**D**) show untreated normal trophozoites and (**D**) a normal schizont. Images (**E**–**I**) show TpcC-treated trophozoites (post 6 h treatment) and (**J**–**L**) show TpcC-treated schizonts (post 6 h treatment). The arrows in (**I**,**H**) highlights the of dark irregular elongated structures, which could indicate hemozoin crystals, within or close to the nuclear material mass. Each image shows a single erythrocyte, with an average diameter of 7.5 μm. Refer to Figure 4 more detailed magnification scales.

**Figure 4 antibiotics-11-00801-f004:**
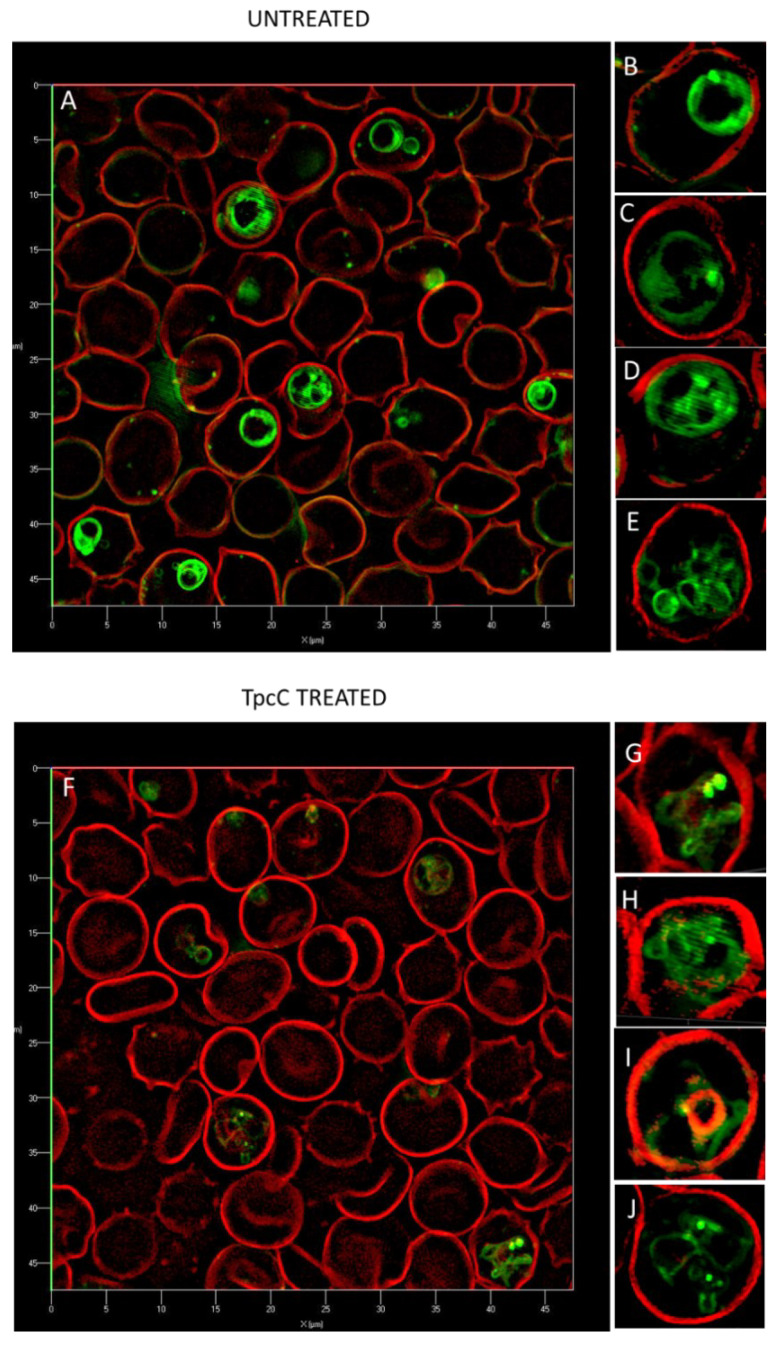
Super-resolution structured illumination fluorescence microscopy images of late intra-erythrocytic stages of *P. falciparum* D10 stained with trypan blue (red) and neutral lipid binding fluorescent dye LipidTOX (green). The large panels are compiled images from super position of both dye images, with four selected enlarged infected cell images are shown on the right. Images (**A**–**E**) show the control cultures (no peptide added) and images (**F**–**J**) show trophozoite stage cultures treated with TpcC after 5 h in culture.

**Table 1 antibiotics-11-00801-t001:** Differences in susceptibility of three strains of *P. falciparum* to Trc analogues with GS as reference peptide. All IC_50_ values are the means of multiple the biological repeats (n) each conducted with triplicate of quadruplicate technical repeats. The resistance index is given as a ratio of activity against CQ resistant strain Dd2 versus CQ sensitive strains, 3D7 and D10. Subscript letters indicate statistically significant differences (Student *t*-test) between activity against different *P. falciparum* strains for each compound. Subscript letters indicate statistically significant differences (One-way Anova with Bonferroni post-test) between activity of different compounds (excluding GS) against a specific strain of *P. falciparum* strains.

Compounds Tested	Toxicity (LC_50_, μM (n)); HC_50_, μM ± SEM (n)		Parasite Strains (IC_50_, nM ± SEM (n))			Resistance Index [23]	Selectivity Index
	COS Cells	Erythrocytes	CQS *P. falciparum* 3D7	CQI *P. falciparum* D10	CQR *P. falciparum* Dd2	IC_50_Dd2/IC_50_ #	IC_50_/LC_50_ *
**TpcC**	10 (2)	9.1 ± 1.5 (4)	126 ± 28 (6) ^a,b,1,2,3^	42 ± 17 (11) ^a,c^	398 ± 94 (3) ^b,c,5^	3; 9	79; 238; 25
**TrcA**	6 (2)	6. 1± 0.6 (5)	41 ± 12 (6) ^d,1^	119 ± 33 (10) ^e,4^	1802 ± 319 (3) ^d,e,5,6,7^	44; 15	146; 51; 3
**PhcA**	8 (2)	7.1 ± 1.6 (8)	23 ± 7 (3) ^f,2^	52 ± 17 (11) ^g^	511 ± 96 (3) ^f,g,6^	22; 10	384; 154; 16
**GS**	9 (2)	6.2 ± 0.4 (9)	1452 ± 151 (6)	1398 ± 86 (16)	1861 ± 160 (6)	1.3; 1.3	6; 6; 5
**CQ**	na	na	17 ± 5 (4) ^h,i,3^	40 ± 3 (12) ^h,j,l,4^	277 ± 18 (6) ^i,j,7^	16; 7	na

# First value IC_50_Dd2/IC_50_3D7, second value IC_50_Dd2/IC_50_D10; * First value IC_50_3D7/LC_50_ (COS cells), second value IC_50_D10/IC_50_ (COS cells); third value IC_50_Dd2/LC_50_ (COS cells).

**Table 2 antibiotics-11-00801-t002:** Summary of in vitro interaction between chloroquine and Trc A, Phc A and Tpc C in different combinations towards CQI *P. falciparum* D10. FICs in terms of intra-erythrocytic *P. falciparum* growth inhibition was determined by the Malstat assay and calculated FIC indices were obtained from three biological repeats of experiments carried out in triplicate.

Peptide	Combination Ratio CQ:Peptide at IC_50_	CQ FIC	Peptide FIC	CQ:Peptide FIC Index *	Isobologram
**Tryptocidine C**	1:2	0.36 ± 0.07	0.90 ± 0.40	1.26 ± 0.36	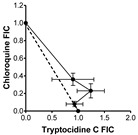
	1:5	0.23 ± 0.08	1.24 ± 0.26	1.47 ± 0.28	
	1:10	0.08 ± 0.03	0.93 ± 0.16	1.02 ± 0.17	
**Tyrocidine A**	1:1	1.77 ± 0.29	1.99 ± 0.36	** 3.76 ± 0.65 **	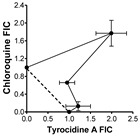
	1:2	0.66 ± 0.02	0.95 ± 0.18	1.60 ± 0.20	
	1:10	0.13 ± 0.10	1.21 ± 0.29	1.35 ± 0.37	
**Phenycidine A**	1:2	1.09 ± 0.03	2.44 ± 0.20	** 3.53 ± 0.22 **	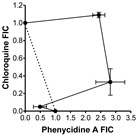
	1:5	0.33 ± 0.15	2.82 ± 0.48	** 3.15 ± 0.33 **	
	1:10	0.05 ± 0.02	0.49 ± 0.22	** *0.54 ± 0.23* **	

* Light-grey cell indicates slight antagonism, dark-grey cell indicates overt antagonism, white cell with normal font indicates sum of action, and white cell with bold italic font indicates synergism.

## Data Availability

Raw data available from authors.

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
