# Peer review of "Antiplasmodial Cyclodecapeptides from Tyrothricin Share a Target with Chloroquine"

_antibiotics, 2022, doi:10.3390/antibiotics11060801_

Round 1

Reviewer 1 Report

Antimicrobial peptides, such as tyrothricin and variants, present a potential as antiparasitic agents for plasmodia. Here, Leussa and Rautenbach (antibiotics-1732200) have studied in details the in vitro antiparasitic activity of tyrocidine A trcA, tryptocidine C TpcC  and phenycidine A (PhcA). The degree of antiparasitic activity (IC50) of tyrocidine A trcA, tryptocidine C TpcC  and phenycidine A (PhcA) assayed in trophozoites of P. falciparum after 48 h of drug exposure is compelling. However, at least authors should address the antiparasitic activity of tyrocidine A or phenycidine A in ring stages of D10 P. falciparum after 72 h of incubation. Authors are recommended to determine the selectivity index (CC50 COS cells/ IC50) and display it in Table 1. In table 2, it is not clear why authors decided to use the combination ratio CQ:peptide using a large excess of trcA, TpcC or PhcA? These peptides are active in low nanomolar range, therefore, authors should tested at 1:1 molar ratio of CQ:peptide. Authors did not specify the incubation time of the drugs with uRBC prior determining the % of hemolysis. Typically, 1 h drug exposure is recommended. A minor comment is that the introduction section is too long and can be edited to become more concise and summarize the main background findings.

Author Response

Reviewer 1: Changes highlighted in yellow in the manuscript

Antimicrobial peptides, such as tyrothricin and variants, present a potential as antiparasitic agents for plasmodia. Here, Luessa and Rautenbach (antibiotics-1732200) have studied in details the in vitro antiparasitic activity of tyrocidine A trcA, tryptocidine C TpcC  and phenycidine A (PhcA). The degree of antiparasitic activity (IC50) of tyrocidine A trcA, tryptocidine C TpcC  and phenycidine A (PhcA) assayed in trophozoites of P. falciparum after 48 h of drug exposure is compelling.

Thank you for the time spent on the manuscript, critical assessment and valuable suggestions.

Query: However, at least authors should address the antiparasitic activity of tyrocidine A or phenycidine A in ring stages of D10 P. falciparum after 72 h of incubation.

Response: In  a previous article by Rautenbach et al. (2007) it was show that tyrocidine A does not have a major effect against ring stage parasites in the 3D7 stain. In this study tryptocidine C (Fig. 2), as well as the tyrocidine mixture containing all the tyrocidines (results now shown in Fig 2) allowed the progression of ring stage parasites to trophozoite stage in a similar time frame as the control (24 hours). We have included a statement on the activity against ring stage parasites, referring to previous results and our results.

It is a good idea testing after 72 hour incubation, but normally P. falciparum cultures need to be fed with fresh erythrocytes and media every 48 hours, depending on parasitaemia. In our assay format 72 hour incubation could lead to false activity, particularly because the control parasitaemia goes up to over 5% after 48 hours.  Also, the time frame of manuscript revision of 10 days and the fact that we lost the original IQR D10 strain and we need to generate intermediate resistant D10 strain to repeat the work in a reasonable time frame. Hopefully our argument above on the lack of activity towards the ring stage is satisfactory.

Query: Authors are recommended to determine the selectivity index (LC50 COS cells/ IC50) and display it in Table 1.

Response: Thank you for the suggestion. The requested information is included in Table 1.

Query: In table 2, it is not clear why authors decided to use the combination ratio CQ:peptide using a large excess of trcA, TpcC or PhcA? These peptides are active in low nanomolar range, therefore, authors should tested at 1:1 molar ratio of CQ:peptide.

Thank you for pointing this problem out – the ratios that were given in the original manuscript were incorrect. We have now changed it to molar ratios at the respective IC50 values in the combinations. We only determined a 1:1 ratio for TrcA, but we did determine a 1:2 ratio for all three peptides. The time frame of manuscript revision of 10 days and the fact that we lost the original IQR D10 strain and we need to generate an intermediate resistant D10 strain, do not allow us to repeat this work in a reasonable time frame. We, however, hope that the reviewer agrees that the current data still support our argument without the 1:1 ratio data for TpcC and PhcA.

Query: Authors did not specify the incubation time of the drugs with uRBC prior determining the % of hemolysis. Typically, 1 h drug exposure is recommended.

Response: Our apology for this omission. We indeed used one hour incubation and have corrected the methodology and included the relevant information.

Query: A minor comment is that the introduction section is too long and can be edited to become more concise and summarize the main background findings.

Response: We worked on the introduction to make it more concise and readable. We hope it fulfils the criteria. 

Reviewer 2 Report

The aim of the manuscript is not clear since some of the authors results are repeated from previous works.  Authors should try to emphasis the difference and importance of their new work in results, and conclusions. I do believe that the title is relevant to the manuscript and very informative. The references of the article relevant and the introductory should be improved in terms of emphasize the difference of the new study in comparison with the published results. Additionally, the article is correctly referenced and appropriate key studies from the past are included. The scientific process is clear but the methods of the study are well presented.  The data is presented in an appropriate way, alongside with the figures and tables. All titles, tables, figures are labelled correctly and the text in the results puts additional value to the data.

To sum up:

  1. Authors should try to emphasis the difference and importance of their new work in results, and conclusions. Specially the conclusions, should mention the new results.
  2. The introductory should be improved in terms of emphasize the difference of the new study in comparison with the published results.

I do think that the manuscript should be published in this journal with some minor revision.

Best regards,

Author Response

Reviewer 2: Changes highlighted in grey

The aim of the manuscript is not clear since some of the authors results are repeated from previous works.  Authors should try to emphasis the difference and importance of their new work in results, and conclusions. I do believe that the title is relevant to the manuscript and very informative. The references of the article relevant and the introductory should be improved in terms of emphasize the difference of the new study in comparison with the published results. Additionally, the article is correctly referenced and appropriate key studies from the past are included. The scientific process is clear but the methods of the study are well presented.  The data is presented in an appropriate way, alongside with the figures and tables. All titles, tables, figures are labelled correctly and the text in the results puts additional value to the data.

Response: Thank you for time spend on our manuscript and the valuable suggestions!

To sum up:

Query: Authors should try to emphasis the difference and importance of their new work in results, and conclusions. Specially the conclusions, should mention the new results.

Response: Thank you for the suggestion. We have worked on the results, discussion and conclusions, as well as the abstract to emphasise the importance and new results of our study.

Query: The introductory should be improved in terms of emphasize the difference of the new study in comparison with the published results.

Response: We worked on the introduction to make it more concise and readable, as well as emphasize the additional new work done in this study. We hope it fulfils the criteria

Round 2

Reviewer 1 Report

none.